# Volatile Organic Compound Composition and Glandular Trichome Characteristics of In Vitro Propagated *Clinopodium pulegium* (Rochel) Bräuchler: Effect of Carbon Source

**DOI:** 10.3390/plants11020198

**Published:** 2022-01-13

**Authors:** Dragana Stojičić, Svetlana Tošić, Gordana Stojanović, Bojan Zlatković, Snežana Jovanović, Snežana Budimir, Branka Uzelac

**Affiliations:** 1Department of Biology and Ecology, Faculty of Sciences and Mathematics, University of Niš, Višegradska 33, 18000 Niche, Serbia; dragana.stojicic@pmf.edu.rs (D.S.); svetlana.tosic@pmf.edu.rs (S.T.); bojan.zlatkovic@pmf.edu.rs (B.Z.); 2Department of Chemistry, Faculty of Sciences and Mathematics, University of Niš, Višegradska 33, 18000 Niche, Serbia; gordana.stojanovic@pmf.edu.rs (G.S.); snezana.jovanovic@pmf.edu.rs (S.J.); 3Department of Plant Physiology, Institute for Biological Research “Siniša Stanković”, National Institute of the Republic of Serbia, University of Belgrade, Bulevar Despota Stefana 142, 11060 Belgrade, Serbia; budimir@ibiss.bg.ac.rs

**Keywords:** carbohydrates, *Clinopodium pulegium*, GC-MS, glandular trichomes, headspace, Lamiaceae, micropropagation, terpenoids, VOC profile

## Abstract

*Clinopodium pulegium* (Rochel) Bräuchler (Lamiaceae) is an endangered species endemic to the Southern Carpathians. It is characterized by the production of high amounts of essential oils, which emit volatile organic compounds (VOCs) that have an essential role in biotic and abiotic stress responses and in plant–plant and plant–insect interactions. The present study was initiated to phytochemically examine the influence of different carbon sources in the nutrition medium on VOC emissions of micropropagated *C. pulegium* plants, using gas chromatography–mass spectrometry analysis of headspace VOCs. The volatile profiles were subjected to multivariate analysis with respect to the presence, concentration and type of carbon source in the nutrient medium. In addition, the effect of different carbohydrates on the density and size of the leaf glandular trichomes, the main structures involved in the emission of VOCs, was determined. A total of 19 VOCs, primarily belonging to mono- and sesquiterpenes previously described in plants, were tentatively identified. Six VOCs were produced at levels higher than 2% of the total VOC emission, dominated by pulegone, ß-pinene and menthone. Inclusion of the carbohydrates in the culture media affected the production of the main leaf trichome-associated volatile allelochemicals although the qualitative composition of the volatiles changed only slightly. Multivariate analysis showed that the concentration, rather than the carbohydrate type, influenced the VOC profile.

## 1. Introduction

Many of the species belonging to the Lamiaceae family are considered aromatic and medicinal plants due to the presence of glandular trichomes that are the main sites for the synthesis of natural bioactive compounds. These bioactive compounds play a crucial role in mediating the plant–environment interactions, often rendering a commercial value to the plants that produce them [1].

The genus *Clinopodium* L. (Lamiaceae) comprises 135 perennial herbs [2], including taxa that were recently transferred from the polyphyletic genus *Micromeria* sect. *Pseudomelissa* based on molecular and morphological/anatomical evidence [3]. Micromorphological traits support the transfer of *Micromeria* species from the section *Pseudomelissa* to the genus *Clinopodium* [4].

*Clinopodium pulegium* (Rochel) Bräuchler (syn. *Micromeria pulegium* (Rochel) Benth.) is a perennial, medium-sized plant, endemic to the Southern Carpathians, with a scattered native area in southwestern Romania and a single known locality in eastern Serbia. It inhabits screes and sheltered calcareous rocks, mainly in the gorges, at altitudes of 1000–1200 m. The plant is woody at the base, with an erect tomentose stem, dark green opposite leaves, and white or lilac-flowered blossoms. The green parts of the plant are densely covered with non-glandular and glandular trichomes.

There are a few available contributions regarding the morphological and phytochemical characterization of the species. The micromorphological investigations of the vegetative and reproductive organs concerned trichome morphotypes, their distribution and abundance [4,5], and one study focused on the morphoanatomical and histochemical analyses of leaf glandular trichomes [6].

In the phytochemical field, there were several contributions related to the analysis of the essential oil composition, mainly of the plants from their native ranges [5,6,7,8] or those cultured in vitro [6]. The profiles obtained by different authors were dominated by the monoterpenic fraction, with the prevalence of oxygenated monoterpenes of the menthane type, although the major compounds differed. Pulegone and menthone represented the main constituents in most of the works [6,8], whereas some authors reported piperitenone oxide [5] or isomenthone [7] as the main compounds. The observed variations in the essential oil composition may be due to different phenological stages, extraction type and variation in the climatic conditions in the studied years. In addition, the chemical composition of extracts of *C. pulegium* plants was determined using solvents of different polarities, related to their antioxidative and antimicrobial potential, as well as phenolic compound content [5,9].

*Clinopodium pulegium* is characterized by the production of high amounts of essential oils which emit the volatile organic compounds (VOCs) that play a crucial role in biotic and abiotic stress responses and in plant–plant and plant–insect interactions. The type and rate of VOC emissions are highly species specific. However, there are no studies focused on the profile of emitted VOCs in *C. pulegium*.

Headspace sampling methods revealed that the most diverse mixture with the highest abundance of VOCs is released predominantly from the flowers in the majority of flowering plants [10]. However, in aromatic and medicinal plants, the leaves are the main source of VOC emission. VOC production and emission are not only organ-specific, but are also regulated developmentally [8,11,12,13,14], and affected by diverse environmental factors [15,16].

In vitro propagation is a useful approach to study the metabolism, differentiation and morphogenesis of plant cells. Exogenous carbohydrates play a key role in the organogenic process during in vitro shoot development. Carbohydrates serve as carbon and energy sources, as their continuous availability is required for shoot primordium growth and/or development [17]. They also act as osmotic agents due to their efficient uptake across the plasma membrane [18]. Carbohydrates are essential to the fundamental processes required for plant growth and development, requiring that their metabolism is carefully coordinated with photosynthate availability, environmental cues, and timing of key developmental programs [19]. Additionally, they can act as signaling molecules that regulate gene expression, thus translating nutrient status to modulation of critical growth processes, to potentially coordinate developmental programs with available carbohydrate [20,21]. Cultured plant tissues require a continuous supply of carbohydrates from the medium, since their photosynthetic activity is reduced due to low light intensity, limited gas exchange and high relative humidity [22]. The morphogenetic and metabolic potential of plant tissues in micropropagation systems can be greatly manipulated by varying the type and the concentration of carbon sources [23].

The large scale production of bioactive compounds of high economic value, with possible future applications in crop pest and weed control, through in vitro culture, requires a comprehensive understanding of the influence that culture conditions have on metabolite production. The present study was initiated to phytochemically examine the influence of different carbon sources in the nutrition medium on the VOC emissions of micropropagated *C. pulegium* plants, using headspace (HS) sample analysis. The VOC profiles were analyzed in relation to the presence, type, and concentration of carbon sources in the nutrient medium. In addition, the effect of different carbohydrates on the density and size of leaf glandular trichomes, the main structures involved in the emission of VOCs, was analyzed.

## 2. Results and Discussion

### 2.1. Effect of Carbon Source on Growth and Biomass Production

The growth and multiplication of *C. pulegium* was obtained on control medium, devoid of carbohydrates, where only about 50% of nodal explants produced 1–2 shortened shoots. Shoots grown on control medium (lacking any carbon source) had a very low proliferation rate (Figure 1A,B) and biomass production (Figure 1D).

Inclusion of different carbohydrates (fructose, glucose or sucrose) in the media had a significant (*p* ≤ 0.05) effect on microshoot formation (Figure 1). The multiplication rate on fructose-supplemented media progressively decreased with an increase of fructose concentration in the culture medium, with a drastic decrease in explant responsiveness observed for 5% fructose (Figure 1A). Microshoot formation on glucose-supplemented media exhibited dynamic changes, including a relatively high frequency of shoot induction even at the highest tested concentration. On sucrose-supplemented media, proliferation was fairly similar at the concentrations up to 4%, but it drastically decreased at the highest tested concentration (5%), compared to lower sucrose concentrations.

The uptake of carbohydrates from the culture medium at the beginning of the culture period is important for the start of shoot induction, as well as their growth and development [24,25]. They are very important energy sources for explants that are not autotrophic in vitro, especially at the initiation stage [25]. The carbohydrate requirements can vary greatly between species and explants, depending on their capacity to absorb and metabolize the molecules of the specific carbon source [26].

Sucrose is considered the best carbohydrate for the supplementation of culture medium in most plants propagated in vitro, due to its efficient uptake through the plasma membrane [27]. Sucrose proved to be the most potent carbohydrate in developing *Digitalis lanata* microshoots, whereas fructose was much less efficient in shoot induction and growth, compared to sucrose or glucose [28]. For each tested carbohydrate type, the optimal shoot regeneration was achieved at the concentration of 3%. Higher concentrations were less productive, whereas sugars at a 2% level had little or no effect on *D. lanata* shoot regeneration.

While sucrose is the most commonly used carbohydrate for plant tissue cultures as the sole carbon and energy source, monosaccharide hexoses (glucose, fructose, or maltose) have been reported to result in better growth in some species [25,29]. Gemas and Bessa [24] tested six types of carbohydrates (sucrose, maltose, glucose, fructose, galactose and sorbitol) and showed that the inclusion of a carbohydrate source was essential for shoot development in *Anacardium occidentale*. The concentration of 83 mM (corresponding to ~1.5% for glucose, fructose, galactose, and sorbitol or ~2.8% for sucrose, and maltose) was sufficient for shoot development in the tested conditions since no significant differences were found when explants were cultured at a higher concentration. Their results showed that regardless of the concentration, it was the type of carbohydrate that influenced shoot response [24].

The highest frequency of shoot induction (93.3%) and mean length of microshoots (24.37 mm) in *C. pulegium* were recorded for explants cultured on medium supplemented with 1% fructose, both following the same trend: Fru > Glu > Suc (Figure 1A,C). The highest number of shoots per explant (>7) was obtained on medium supplemented with fructose and sucrose, at a slightly higher concentration (2%; Figure 1B). The highest explant biomass was recorded on medium supplemented with either carbon source at the concentration of 1% (Figure 1D). In general, the highest proliferation was achieved on media supplemented with lower carbohydrate concentrations, with fructose being the most beneficial for shoot multiplication and elongation. For that reason, *C. pulegium* plants grown at lower carbohydrate concentrations (1% and 2%) were selected for further analyses.

### 2.2. Effect of Carbon Source on C. pulegium VOC Profiles

The VOC emission profiles of in vitro cultured *C. pulegium* revealed a total of 19 compounds, out of which 17 were common for all examined samples (Table 1). The terpenoid fraction represented up to 99.8% of the total VOC profile, unevenly distributed between mono- and sesquiterpenoids (Figure 2).

The VOC profiles of in vitro grown *C. pulegium* were dominated by oxygenated monoterpenes, varying from 53.3% (Suc 2%) to 72.8% (Fru 2%). These were followed by monoterpene hydrocarbons (summing up to 41.6%), while sesquiterpene hydrocarbons had the lowest relative abundance (0.7–5.2%). Terpenoids account for the largest share of the volatile organic compounds in the majority of examined Lamiaceae species, including those belonging to the genus *Clinopodium*. In all investigated *Clinopodium* species, the main volatile constituents belonged to oxygenated monoterpenes of the menthane type [5,6,7,8,14,30,31].

The main compound in the VOC profile of plants grown on medium devoid of carbohydrates was menthone (*8*, 47.1%), followed by ß-pinene (*4*, 19.8%), isomenthone (*9*, 12.3%), and pulegone (*12*, 8.5%) (Table 1). The relative abundance of α-pinene (*2*, 4.1%) and limonene (*6*, 2.7%) was within the range 5.0–2.0%, while *cis*-piperitone oxide (*13*, 1.7%) and sabinene (*3*, 1.5%) were in the range 2.0–1.0%. The remaining compounds occurred at percentages lower than 1.0% or in traces (<0.05%). Plants grown on medium devoid of carbohydrates contained one exclusive compound, isomenthyl acetate (*14*, 0.1%).

The inclusion of carbohydrates in the culture media affected the production of the main volatiles although the qualitative composition of the VOC emissions changed only slightly. The VOC profiles of plants grown on medium supplemented with carbohydrates were dominated by pulegone (*12*, 34.6%–53.4%), ß-pinene (*4*, 12.1%–26.5%), and menthone (*8*, 9.2%–17.6%), as shown in Table 1. Generally, in these plants, compounds showing relative abundance in the range 5.0–2.0% were isomenthone (*9*), α-pinene (*2*), and limonene (*6*). ß-myrcene (*5*) was in the range 2.0–1.0%, while the percentage range of sabinene (*3*) and germacrene D (*17*) varied with the carbohydrate type and concentration. The remaining compounds generally occurred at percentages lower than 1.0% or in traces (<0.05%). Plants cultured in the presence of carbohydrates contained (*3Z*)-hexenol (*1*) as an exclusive compound, although found in very low to trace amounts.

VOC production and emission depend on the type and ontological stage of a particular organ [1,11,32,33,34]. In a number of Lamiaceae species, the foliar VOC profile is more complex than the floral one, as evidenced by the presence of a higher number of both total and exclusive compounds [35,36,37]. Due to the more complex VOC profile, dominated by defensive compounds, the leaves were primarily ascribed a protective action. In contrast, the most abundant compounds exclusive to the floral profile, underlined the major attractant role for the reproductive organs [36]. The main compounds detected in all *C. pulegium* VOC profiles in this study: pulegone (*12*), ß-pinene (*4*) and menthone (*8*) are suggested to have the ecological role in defense against diverse environmental challenges [38,39,40]. Therefore, the VOC profiles of in vitro cultured *C. pulegium* indicated a marked repellent action at the vegetative organ level, primarily leaf.

Carbohydrates affected the ratio between the two monoterpene fractions, with sucrose exerting the most prominent effect (Table 1). Compared to control plants, the relative percentage of pulegone in plants cultured in the presence of carbohydrates was inversely correlated with that of menthone and isomenthone. Furthermore, a higher sesquiterpene fraction (lower monoterpene-to-sesquiterpene ratio) was also observed in *C. pulegium* plants cultured in the presence of carbohydrates, especially at higher sugar concentrations, compared to control plants (Figure 2). Sesquiterpenic compounds, which are typically produced in a number of plant species for their antimicrobial and insecticidal properties, as well as their role as insect pheromones [1,37], also display various pharmacological effects and are associated with diverse health benefits [41].

VOC production and emission of in vitro cultures are affected by environmental factors such as light and temperature, micropropagation techniques, nutrient medium composition, plant growth regulators [6,31,42,43]. Stojičić et al. [6] showed that micropropagated *C. pulegium* plants differed phytochemically from their parent plants from natural populations, and that their secondary metabolite production was significantly altered in response to plant growth regulators in the culture medium. In this study, the type and the concentration of carbohydrates affected not only growth and development of *C. pulegium* plants, but also their VOC profiles, suggesting that the careful selection of the culture conditions could increase their production of secondary metabolites.

### 2.3. Multivariate Analyses (CDA and CA) of C. pulegium Volatiles

The VOC profiles of 210 *C. pulegium* plants cultured on media supplemented with 1% or 2% fructose (F1, F2), glucose (G1, G2) or sucrose (S1, S2) and control group (C) were compared by canonical discriminant analysis (CDA) and by cluster analysis (CA). Analyses were performed with non-transformed values, which depict the natural variability of the analyzed characters (VOCs).

The CDA with seven groups of *C. pulegium* plants differing in their carbohydrate treatments, considered as seven a priori groups, showed that the first two canonical axes participated in 84.0% of the total discrimination, of which the first axis (CA 1) accounted for 70.0% (Table 2, Figure 3). The scatter plot in the projection of the first two axes revealed a clear separation of the control group (C) situated in the negative zone of CA 1, compared to a composite group of plants grown on sugar-supplemented media. Five compounds, mostly dominant volatiles of examined *C. pulegium* groups, had a significant impact on both axes (ß-pinene, menthone, pulegone, (*E*)-caryophyllene, and sabinene). Isomenthone considerably affected only the CA 1, while limonene and *trans*-calamenene considerably affected only the CA 2 (Table 2). The values of the coefficients by which ß-pinene, menthone, isomenthone, pulegone, (*E*)-caryophyllene, and sabinene primarily burden the first canonical axis, bore about 70% of the overall variability, thus providing an absolute contribution to the discrimination (Table 2). This indicated the dissimilarity of HS volatiles between the control plants and those exposed to exogenous carbohydrates. The mean values of menthone, ß-pinene and isomenthone were higher, while the values of pulegone, sabinene and (*E*)-caryophyllene were relatively lower in the control group. Further segregation of plants cultured on media containing fructose (F1, F2), glucose (G1, G2) and sucrose (S1, S2), comprising a common group in the graph, was not clear due to a contrast when compared to control group. The cloud of those individuals is mainly located in the positive zone, relative to the first canonical axis (Figure 3). The exclusion from this group referred to the segregation of S1 which slightly penetrated the negative zone of the CA 1.

Additionally, cluster analysis showed that the plants grown on fructose- or glucose-supplemented media were the most similar in volatiles to each other, while both groups grown on sucrose exhibited some degree of individuality (Figure 4). The greatest difference was observed for the control group, which showed a low level of similarity to all other groups. However, these results are sufficient to assume the existence of two clear entities: individuals grown on medium that does not contain carbohydrates (C) and those belonging to groups F1, F2, G1, G2, S1 and S2, grown on media supplemented with sugars.

The CDA of HS volatiles derived from the groups of plants grown exclusively on carbohydrate-supplemented media provided a clearer picture of their segregation. The first discriminant axis in this case participated in 55.5% of the total discrimination, and the second axis with 21%, which was sufficient to describe the overall sample variability (Table 3). Groups of plants grown at lower (1%) carbohydrate concentration differed from the groups grown at higher (2%) concentrations, as separated along the first canonical axis. Both sucrose groups slightly differed from each other and from the rest of the groups, also in relation to the first canonical axis (Figure 5). The segregation along CA 1 is approximately determined by the content of *trans*-calamenene, limonene and (*E*)-caryophyllene (Table 3). The second discriminant axis, which was mostly defined by the content of sabinene, (*E*)-caryophyllene, β-myrcene, isomenthone, bicyclogermacrene, and α-pinene (Table 3), generally separated sucrose groups (S1, S2) from all monosaccharide groups (Figure 5). The cluster analysis in this case also confirmed the results presented in the CDA (Figure 6). The groups consisting of individuals grown at lower glucose and fructose concentrations and those cultured at higher concentrations differed from each other, but still displayed similarity when compared to the volatile profiles of plants grown on sucrose. Plants grown at lower sucrose concentration and those grown at higher sucrose concentration somewhat differed in their volatile profiles from other groups, as well as from each other. These results were sufficient to assume the following: (1) plants grown on media containing carbohydrates at 1% (G1, F1, S1) were generally different in volatile composition from those grown at higher concentrations (G2, F2, S2) and (2) to a lesser degree, plants grown on media containing different types of carbohydrates also differed in their volatile profiles. This difference was more prominent in plants cultured on media containing sucrose (S1, S2).

Based on the obtained results, it is apparent that the presence, as well as the concentration of carbohydrates in culture media, significantly affected the composition of HS volatiles in *C. pulegium*. Differences in the composition of volatile profiles were evident even in plants grown on media that differed in the type of sugar used in their design. On the one hand, different concentrations of a certain carbohydrate type were shown to clearly affect the volatile chemical composition (as was the case with S1, S2), while in the case of monosaccharides the sugar concentration did not affect the qualitative composition of the volatiles to such a great extent.

### 2.4. Effect of Carbon Source on Glandular Trichome Characteristics

The leaves of in vitro grown *C. pulegium* displayed an indumentum consisting of non-glandular and glandular trichomes, the latter including both peltate and capitate types as a characteristic feature of Lamiaceae. Two main glandular trichome morphotypes were present on both adaxial and abaxial surfaces, and showed an apparently random distribution over the entire leaf surface (Figure 7A,B and Figure 8).

Peltate trichomes of *C. pulegium* (Figure 7C–E) were more abundant on the abaxial leaf side (Figure 8B). Two submorphotypes of capitate trichomes were identified in this plant species. Although found in only a few Lamiaceae species, type I capitate trichomes (C1) were by far the most abundant morphotype present on leaves of in vitro cultured *C. pulegium*, and were more abundant on the abaxial side (Figure 7F and Figure 8C). Type II capitate trichomes (C2), positioned perpendicular to the leaf surface, appeared more frequently on the adaxial leaf surface (Figure 7G–I and Figure 8D). Stojičić et al. [6] showed that the secretion of all glandular trichome morphotypes of in vitro grown *C. pulegium* was stained intensively only in response to lipophilic dyes. The observed staining susceptibility to NADI reagent and Nile blue A indicated that trichome secretion was almost exclusively composed of terpenes, which comprise the majority of VOCs emitted from *C. pulegium* leaves in vitro.

For each glandular trichome morphotype, the number of trichomes per unit leaf area and their size were determined for plants cultured on medium supplemented with 2% fructose, glucose or sucrose. The number of each trichome morphotype per mm^2^ of both epidermis surfaces is shown in Figure 8A. The number of peltate and type II capitate trichomes per mm^2^ was the highest on glucose-supplemented medium, whereas density of type I capitate trichomes was unaffected by the carbohydrate type. Leaf surface (adaxial vs. abaxial) had a significant effect on the density of each glandular trichome morphotype (Figure 8B–D). In addition, plants cultured on glucose-supplemented medium also had the highest number of peltate trichomes per leaf unit area on the abaxial leaf side, whereas on the adaxial side their number was not affected by the type of carbohydrate (Figure 8B). The mean numbers of each type of *C. pulegium* capitate trichomes present on either leaf side did not differ significantly among the treatments (Figure 8C,D).

The total number of *C. pulegium* glandular trichomes (P + C1 + C2) per leaf unit area on the adaxial leaf side did not vary with the treatment, but on the abaxial side the trichome density was significantly increased by glucose (433.3 ± 17.0 mm^−2^), compared to sucrose (335.9 ± 14.6 mm^−2^). Higher density of glandular trichomes on the abaxial relative to the adaxial leaf surface has been reported in a number of aromatic plant species [44,45,46]. Tozin et al. [44] reported an increase in the density of glandular trichomes on the adaxial leaf surface of *Ocimum gratissimum* upon herbivory. Moreover, the authors demonstrated a differential response of each glandular trichome morphotype, consistent with the idea of a compartmentalization of functions among the different trichome morphotypes in the plant defense against environmental factors [44].

Different carbohydrates exerted only minor effects on the average diameter of *C. pulegium* glandular trichomes. The diameter of peltate trichomes was slightly higher in plants cultured on medium supplemented with 2% sucrose, but did not differ significantly between adaxial and abaxial leaf surfaces (Figure 9A,B). An average C1 trichome diameter was slightly higher in plants cultured on media supplemented with 2% glucose or sucrose. The adaxial leaf side was characterized by larger C1 trichomes for each carbohydrate tested, compared to the abaxial side (Figure 9A,C). The diameter of C2 trichome heads was slightly higher on the abaxial leaf side, with an average diameter slightly higher in plants cultured on glucose-supplemented medium (Figure 9A,D).

Glandular trichome density is species- and organ-specific [47]. However, the density of glandular trichomes changes over the course of leaf development. Whether their final number was established early during leaf differentiation [48,49,50] or was not fixed at the time of leaf emergence so that new trichomes were formed throughout all stages of leaf ontogeny [16,34], most studies concluded a progressive decrease in trichome density in the course of leaf development [11,34,48,50]. As a consequence, trichome density in many plant species is very high in young leaves, but decreases rapidly with leaf expansion [11,34,49,50]. In vitro cultured plants are, by definition, considered as juvenile-stage plants. In our study, *C. pulegium* leaves used for the analysis of glandular trichome characteristics were the uppermost, not fully developed young leaves. Therefore, the relatively high trichome density that was observed in this study is in accordance with findings that in young developing organs, which are most vulnerable to predators, glandular trichomes are very densely arranged and active in secretion [32,50].

In addition, trichome density depends on the plant’s environment [51,52]. Given that leaf trichomes are often induced following damage by insect herbivores, Traw and Bergelson [53] investigated the effects of artificial damage, jasmonic acid, salicylic acid, and gibberellin on the induction of trichomes in Arabidopsis. They showed that artificial damage and jasmonic acid caused significant increases in leaf trichome production, while salicylic acid had a negative effect on trichome production and consistently reduced the effect of jasmonic acid. However, Kjær et al. [54] showed that applying stress did not increase the size or number of glandular trichomes in *Artemisia annua*. Stress treatments exerted only a minor promoting effect on the initiation of glandular trichomes in developing leaves, and a maturing effect later in the lifetime of the individual trichomes. On the contrary, trichome density, as well as artemisinin production was shown to increase in leaves and flowers of transgenic *A. annua* plants following *Agrobacterium*-mediated transformation [55].

In tomato, type VI glandular trichomes and their derived volatiles were strongly induced by jasmonic acid [16]. Although observed in all developmental stages, a JA-induced increase in type-VI trichome densities, as well as increase in the concentration of trichome-derived volatiles, was greater in developing leaves than in fully developed tomato leaves. The increase in trichome-derived volatiles in fully developed leaves was explained by an enhanced production per trichome, while in developing leaves this was mainly caused by increases in type-VI trichome densities. The authors thus showed that a JA-mediated induction of trichome density and chemistry in tomato depended on leaf developmental stage, and that it directly resulted from modifications in epidermal cells [16]. JA did not increase the induction of type-VI trichome density in fully developed leaves, but it increased their biosynthetic capacity to produce more volatiles. According to their results, the density of type-VI glandular trichomes was not fixed at the time of leaf emergence, but induction was no longer feasible once the leaves had reached a mature developmental stage.

The response of leaf glandular trichomes to environmental factors is particularly important for medicinal plants, given that bioactive metabolites are synthesized in these structures. Studies concerning the effect of culture conditions on glandular trichome density are scarce. Compared to field- or greenhouse-grown plants, in vitro plants are exposed to high relative humidity in culture vessels, low irradiance, low concentration of CO_2_ during photoperiod and poor aeration, which may lead to the occurrence of abnormalities in plant morphology, anatomy and physiology [56]. Benlarbi et al. [57] showed that the number and the external appearance of glandular (peltate) trichomes of *Mentha pulegium* greatly depended on the growth conditions. In plants cultured in vitro, the peltate trichome number was markedly lower compared to plants grown in pots under greenhouse conditions, and they appeared empty, as opposed to the turgid and swollen trichomes of greenhouse plants. This correlated well with a decreased amount of volatiles produced by the plants cultured in vitro [57]. Concerning culture medium composition, Stanojković et al. [46] showed that the highest leaf trichome density was observed in *Inula britannica* plants grown in vitro on medium devoid of carbohydrates, and that the addition of sucrose to nutrient medium led to a decrease in the number of glandular trichomes per mm^2^ on both leaf surfaces. Trichome density in all treatments was significantly higher on the abaxial side of *I. britannica* leaves.

In this study, a substantial increase in trichome density was observed only in peltate trichomes when plants were cultured in the presence of 2% glucose, where they also reached a maximum diameter. Some reports showed a positive correlation between the density of peltate trichomes and essential oil production in aromatic plants [57,58]. Therefore, the potential use of *C. pulegium* plants in the production of valuable bioactive metabolites might be manipulated by environmental factors that influence the number and secretory activity of their glandular trichomes.

## 3. Materials and Methods

### 3.1. Plant Material

Shoot cultures of *Micromeria pulegium* were established from the aerial parts of randomly collected plants at the vegetative stage of development, harvested in June 2012 from wild-growing populations in Svrljiški Timok gorge (43°32′23″ N, 22°10′16″ E), as described by Tošić et al. [9].

### 3.2. Shoot Multiplication and Biomass Production

Micropropagation was carried out using one-node stem segments (~1 cm) bearing two axillary buds on basal MS medium [59] supplemented with 3% sucrose (*w*/*v*) and 0.7% (*w*/*v*) agar (Torlak, Belgrade), in 370 mL glass jars containing 50 mL of culture medium (ten explants per jar).

The pH of the media was adjusted to 5.8 prior to autoclaving at 114 °C for 25 min. All cultures were exposed to a 16 h photoperiod and maintained under diffuse light provided by cool white fluorescent lamps, with a photon flux density of 45 µmol m^−2^ s^−1^, at 25 ± 2 °C. Routine subculture was performed in 4-week intervals.

Nodal segments derived from shoot cultures, grown on basal medium for four weeks, were transferred to MS medium supplemented with various carbon sources at different concentrations: fructose (Fru), glucose (Glu), or sucrose (Suc) at 1, 2, 3, 4 or 5% (*w*/*v*). Basal MS medium devoid of carbohydrates was used as a control. After 4 weeks in culture, the number of explants producing shoots, the number of shoots per explant and the shoot length, as well as explant fresh weight were recorded in order to evaluate the effect of different carbon sources on shoot multiplication and biomass production.

### 3.3. Scanning Electron Microscopy

For scanning electron microscopy (SEM), fresh leaves isolated from shoots cultured on MS medium containing either carbon source at 2% were used. From each plant, the second pair of leaves below apex was sampled. Leaf samples were coated with a thin layer of gold and palladium in a BAL-TEC SCD 005 sputter coater. Both adaxial and abaxial surfaces, from each side of the midrib, were examined with a JEOL JSM-6390 LV (JEOL, Tokyo, Japan) scanning electron microscope operated at 15 kV.

### 3.4. Headspace Isolation

The headspace sampling conditions were as described by Nikolić et al. [60]. A total of 150 mg (per sample) of fresh plant material, comprising in vitro plantlets collected at the end of 4-week subculture, was used. Two replicates, each consisting of at least 15 randomly chosen plants, were used per treatment. Each sample was immediately inserted into 20 mL headspace vials, to which 1 mL of distilled water was added. The prepared vials were placed into a tray for further automated procedure. Each sample was heated at 80 °C for 20 min with the following program: shaking for 5 s, pausing for 2 s. After equilibration, the gas vapor of headspace volatiles above the plant material was sampled with a gas syringe and subsequently injected into a GC analyzer (sample volume 500 μL; split ratio 10:1).

### 3.5. GC-MS Analysis

Samples were analyzed using a 7890B GC equipped with a fused silica capillary column (HP-5MS, 25 m × 250 μm, film thickness 0.25 μm; Agilent Technologies, Santa Clara, CA, USA) and coupled with a 7000B MS/MS spectrometer (operating in MS1 scan mode) from the same company. The GC was operated under the following conditions: injector temperature 250 °C; GC-MS interface temperature 300 °C; oven temperature programmed from 50 °C for 2.25 min, then to 200 °C at a rate of 4 °C min^−1^ (carrier gas He, flow rate 1.0 mL min^−1^, constant flow mode). MS conditions were as follows: ionization voltage of 70 eV; acquisition mass range 40–440; scan time 0.32 s. HS volatiles were identified from TIC (Total Ion Chromatogram) by comparison of their linear retention indices relative to C_8_-C_20_
*n*-alkanes recorded on the same column/temperature program with literature values (NIST MS Search 2.0; NIST Chemistry WebBook SRD69; Adams [61]) and their mass spectra with those of standards from databases (Wiley 6, NIST02, Adams) by the application of the AMDIS software (the Automated Mass Spectral Deconvolution and Identification System, Ver. 2.7, distributed within the software package for 7890–7000 BGC-MS/MS triple quadrupole system). The percentage composition of the HS volatiles was computed from the Total Ion Chromatogram’s peak areas without any corrections.

### 3.6. Statistical Analysis

The efficiency of different carbon sources on growth was determined after 4 weeks in culture by comparing biomass and in vitro proliferation rates. Biomass increase was calculated on fresh weight basis. Proliferation rate was assessed by counting the number of shoots following 4-week carbohydrate treatment under previously defined conditions. The data obtained from two repeated experiments, with sixty explants per each medium (ten explants per jar) were averaged and statistically analyzed, and differences were tested for significance using ANOVA and Tukey’s HSD test at the significance level of *p* ≤ 0.05.

The effect of different carbon sources on the size and density of different glandular trichome morphotypes was determined after 4 weeks in culture. At least two replicates per each of 6 plants grown on either medium (2% Fru, Glu, or Suc) were examined, and two observation fields for each surface were selected, totaling in 24 observation fields per treatment. For each leaf, the trichome density was expressed as the mean number ± standard error of respective counts per mm^2^. The data obtained from two repeated experiments were averaged and statistically analyzed, and differences were tested for significance using ANOVA and Tukey’s HSD test at the significance level of *p* ≤ 0.05.

The statistical analysis of the HS volatiles data set was performed using the STATISTICA software for Windows ver. 8 (StatSoft, Inc., Tulsa, OK, USA). It included descriptive statistics, canonical discriminant analysis (CDA), and cluster analysis (CA). CDA was performed to test the hypothesis that the analyzed sample was composed of several discrete groups of individuals, which were chemically differentiated from each other. From the total data set of original HS compounds, 19 volatile compounds were selected for multivariate analyses. The volatiles from seven a priori groups of individuals, regarding different concentrations of glucose (G1, G2), fructose (F1, F2) and sucrose (S1, S2) in culture medium, including a control group (C) were processed by CDA. The similarity in the composition of volatile profile was determined by CA, employing linkage distance as a method to determine the distance between groups.

## 4. Conclusions

Our study showed that *C. pulegium* growth and metabolism were influenced by carbohydrates in vitro. Fructose, glucose, and sucrose at lower concentrations (1% and 2%) promoted shoot multiplication and growth, as well as biomass accumulation, indicating higher regeneration and propagation rates of in vitro grown *C. pulegium*. The influence of different carbon sources in the nutrition medium on VOC emissions of micropropagated *C. pulegium* plants was phytochemically examined using gas chromatography–mass spectrometry analysis of headspace VOCs. A total of 19 glandular trichome-derived volatiles, primarily belonging to monoterpenes, dominated by pulegone, ß-pinene, and menthone, were identified. Multivariate analyses showed that the presence, as well as the concentration of carbohydrates in culture media, significantly affected the composition of the HS volatiles in *C. pulegium*, with the concentration effect being more pronounced for sucrose than for monosaccharides. A number of pharmacological and biological effects that have been associated with the main monoterpenoids detected in *C. pulegium* VOCs pave the way to their possible future applications in crop pest and weed control. Therefore, our data suggest that *C. pulegium*, cultured under the favorable in vitro conditions indicated in this work, could be an important source of valuable monoterpene derivatives throughout the year.

## Figures and Tables

**Figure 1 plants-11-00198-f001:**
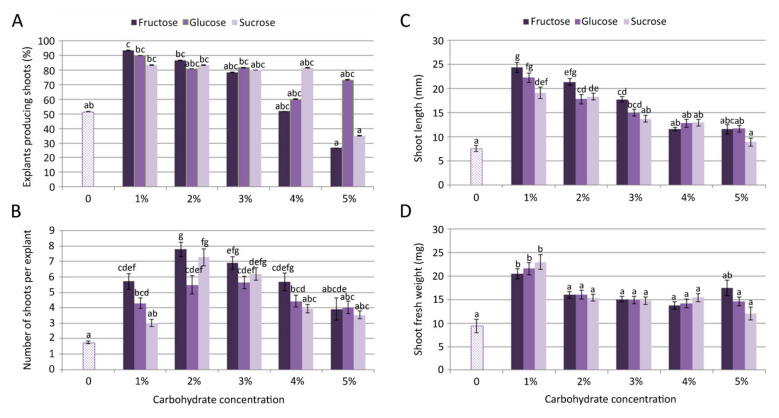
Effect of carbon source on *Clinopodium pulegium* shoot multiplication: frequency of shoot induction (**A**), number of shoots per explants (**B**), shoot length (**C**) and shoot fresh weight (**D**) after 28 days of culture. Values represent means ± SE, *n* = 60. Means within the same histogram marked with different letters are significantly different according to Tukey’s HSD test at *p* ≤ 0.05.

**Figure 2 plants-11-00198-f002:**
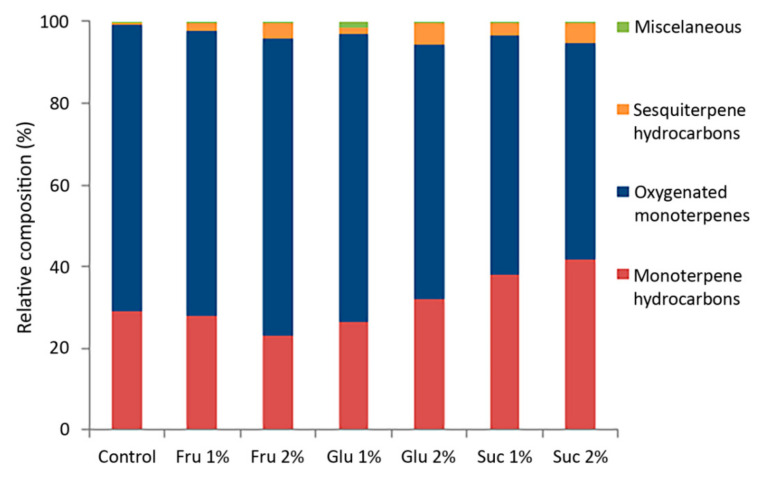
Relative composition (%) of major classes of compounds from in vitro grown *Clinopodium pulegium*, cultured on media supplemented with different carbohydrates.

**Figure 3 plants-11-00198-f003:**
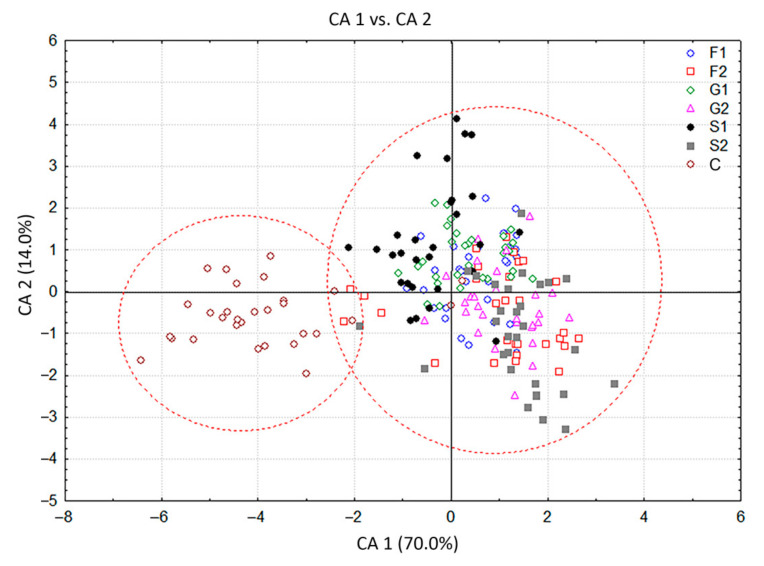
Canonical discriminant analysis (CDA) based on contents of 19 *Clinopodium pulegium* HS volatiles isolated from 210 plants of seven a priori groups.

**Figure 4 plants-11-00198-f004:**
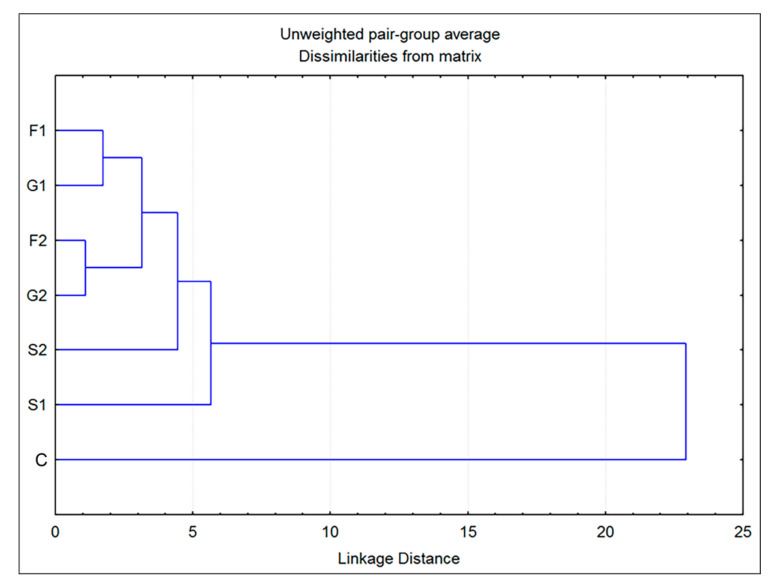
Dendrogram obtained by cluster analysis of 7 groups of *Clinopodium pulegium* plants based on 19 HS volatiles.

**Figure 5 plants-11-00198-f005:**
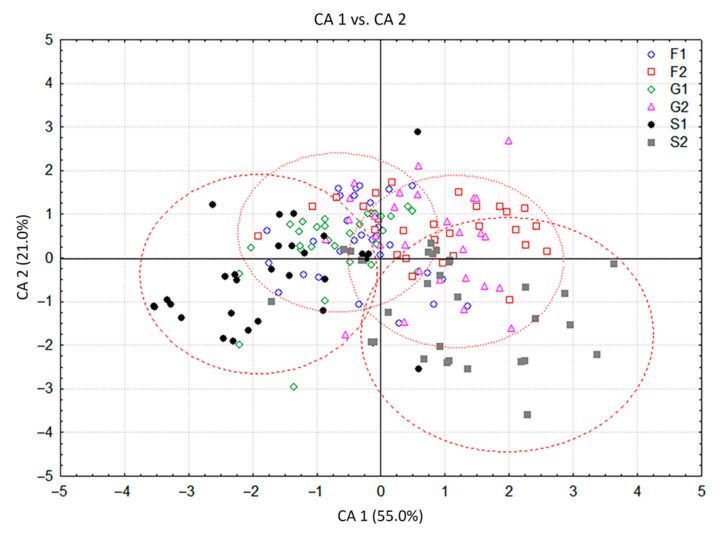
Canonical discriminant analysis (CDA) based on contents of 18 *Clinopodium pulegium* HS volatiles isolated from 180 plants of six a priori groups.

**Figure 6 plants-11-00198-f006:**
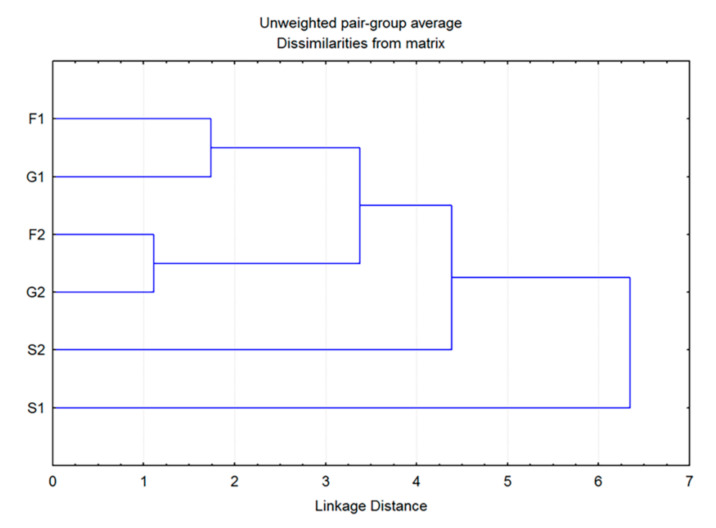
Dendrogram obtained by cluster analysis of 6 groups of *Clinopodium pulegium* plants based on 18 HS volatiles.

**Figure 7 plants-11-00198-f007:**
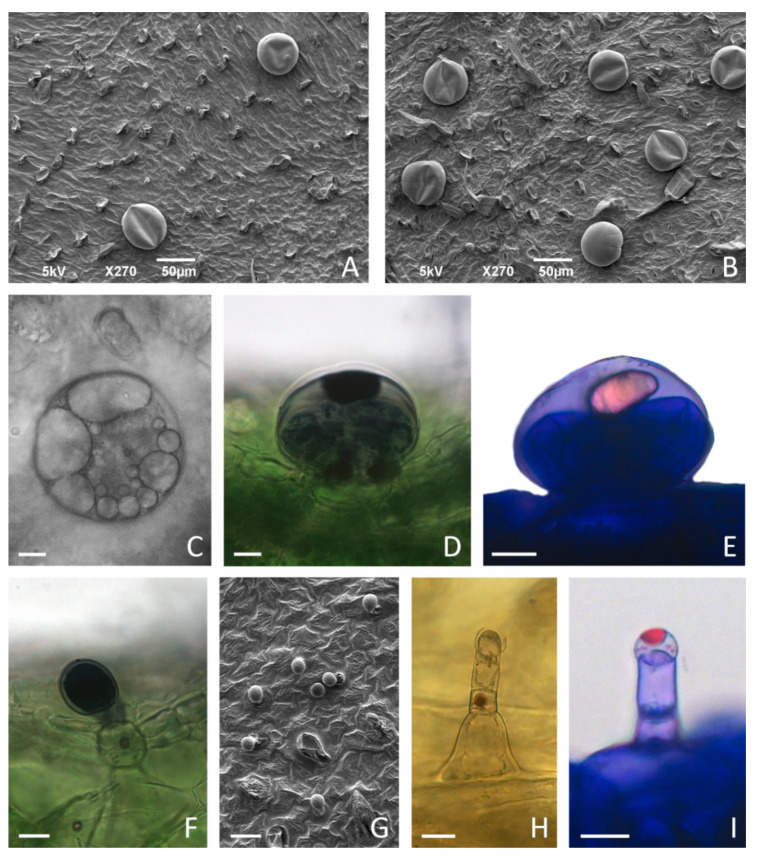
Glandular trichome morphotypes on the leaves of in vitro cultured *Clinopodium pulegium*. SEM micrographs of adaxial (**A**,**G**) and abaxial (**B**) surface of leaves of plants cultured on medium supplemented with 2% sucrose. (**C**–**E**) Peltate trichomes on leaves of plants cultured on medium supplemented with 3% sucrose, consisting of a basal cell embedded in the epidermis, a short unicellular stalk and a large round multicellular head comprising eight secretory cells; large amounts of secretory product present within subcuticular space (**C**) stained positively with NADI reagent for terpenoids (**D**) and with Nile Blue A for essential oils (**E**). (**F**) Procumbent type I capitate trichomes (C1), consisting of a basal cell with a short cutinized stalk and unicellular ellipsoidal head, showed positive NADI reaction of head and stalk cells. (**G**–**I**) Type II capitate trichomes (C2), consisting of a conical basal cell, uni- to bicellular stalk and elongated unicellular head; C2 trichomes are particularly numerous on adaxial leaf surface (**G**), positioned upright (**H**), with apical secretory cell developing subcuticular space filled with secretory droplets positive for essential oils after staining with Nile Blue A (**I**). Scale bars: 50 µm (**A**,**B**); 20 µm (**G**); 10 µm (**C**–**F**), (**H**–**I**).

**Figure 8 plants-11-00198-f008:**
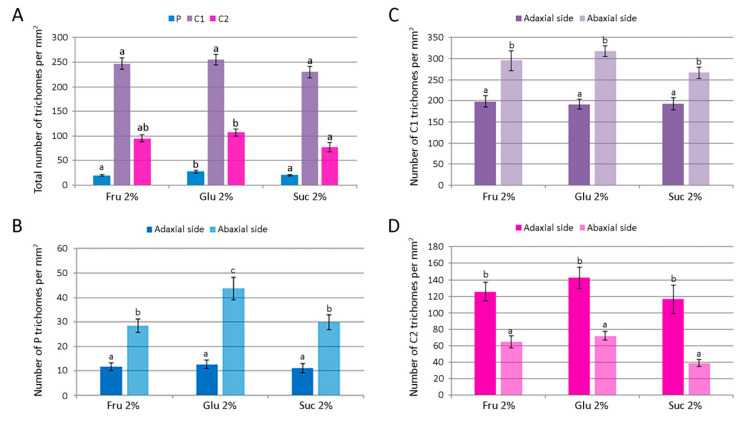
The number of glandular trichomes per unit area of leaves of *Clinopodium pulegium* plants cultured in vitro on medium supplemented with different carbohydrates at the concentration of 2%. (**A**) Overall density of different trichome morphotypes on both epidermis surfaces. (**B**–**D**) Distribution and density of peltate (**B**), type I capitate (**C**) and type II capitate (**D**) trichomes. Values represent means ± SE, *n* = 24. Means within the same histogram marked with different letters are significantly different according to Tukey’s HSD test at *p* ≤ 0.05.

**Figure 9 plants-11-00198-f009:**
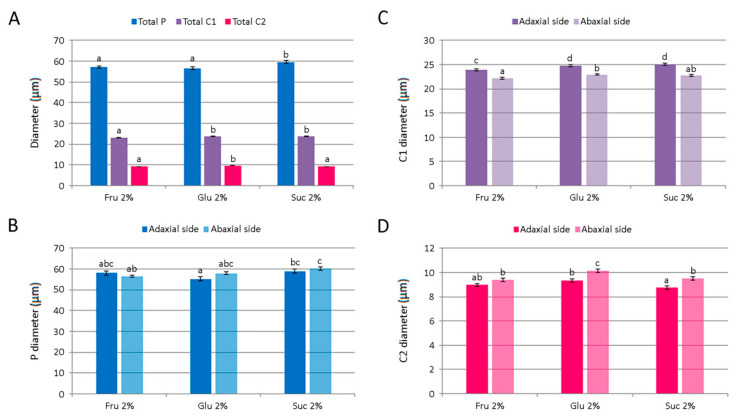
The size of glandular trichomes on leaves of *Clinopodium pulegium* plants cultured in vitro on medium supplemented with different carbohydrates at the concentration of 2%. (**A**) An average diameter of different trichome morphotypes of both epidermis surfaces. (**B**) Diameter of glandular head of peltate trichomes. (**C**) Diameter of the head and neck cell of type I capitate trichomes. (**D**) Diameter of the head cell of type II capitate trichomes. Values represent means ± SE, *n* = 24. Means within the same histogram marked with different letters are significantly different according to Tukey’s HSD test at *p* ≤ 0.05.

**Table 1 plants-11-00198-t001:** Relative percentage composition of HS volatiles of *Clinopodium pulegium* shoot cultures grown on media supplemented with different carbon sources.

				Relative Abundance (%) ^3^						
Entry	Compound	RI ^1^	LI ^2^	Control	Fru 1%	Fru 2%	Glu 1%	Glu 2%	Suc 1%	Suc 2%
*1*	(*3Z*)-Hexenol	852	850	-	0.14 ± 0.01 ^a^	tr	0.07 ± 0.03 ^a^	tr	0.07 ± 0.05 ^a^	tr
Monoterpenoids										
	*Monoterpene hydrocarbons*									
*2*	*α*-Pinene	935	932	4.11 ± 0.34 ^a^	3.96 ± 0.35 ^a^	3.83 ± 0.27 ^a^	3.72 ± 0.29 ^a^	4.85 ± 0.43 ^ab^	4.93 ± 0.44 ^ab^	5.64 ± 0.37 ^b^
*3*	Sabinene	975	969	1.55 ± 0.12 ^a^	1.75 ± 0.11 ^ab^	1.79 ± 0.12 ^ab^	1.92 ± 0.13 ^ab^	2.16 ± 0.21 ^abc^	2.33 ± 0.20 ^bc^	2.64 ± 0.18 ^c^
*4*	** *ß-* ** **Pinene**	978	974	**19.78 ± 1.74 ^abc^**	**17.23 ± 2.18 ^abc^**	**12.10 ± 1.01 ^a^**	**16.61 ± 2.14 ^ab^**	**18.62 ± 2.57 ^abc^**	**26.48 ± 3.69 ^c^**	**25.81 ± 2.31 ^bc^**
*5*	*ß*-Myrcene	992	988	0.73 ± 0.07 ^a^	0.96 ± 0.05 ^ab^	1.39 ± 0.10 ^c^	1.08 ± 0.06 ^bc^	1.39 ± 0.10 ^c^	0.98 ± 0.06 ^ab^	1.34 ± 0.09 ^c^
*6*	Limonene	1030	1024	2.72 ± 0.27 ^a^	4.01 ± 0.52 ^ab^	3.94 ± 0.34 ^ab^	3.02 ± 0.21 ^ab^	4.90 ± 0.63 ^bc^	3.32 ± 0.37 ^ab^	6.15 ± 0.74 ^c^
	*Oxygenated monoterpenes*									
*7*	Isopulegol	1150	1145	tr	0.08 ± 0.03 ^a^	0.18 ± 0.05 ^a^	0.08 ± 0.03 ^a^	0.21 ± 0.07 ^a^	0.22 ± 0.10 ^a^	0.07 ± 0.03 ^a^
*8*	**Menthone**	1156	1148	**47.07 ± 3.10 ^b^**	**14.14 ± 1.36 ^a^**	**11.82 ± 1.60 ^a^**	**14.50 ± 1.41 ^a^**	**9.20 ± 1.16 ^a^**	**17.56 ± 1.78 ^a^**	**12.51 ± 2.36 ^a^**
*9*	Isomenthone	1167	1158	12.32 ± 2.39 ^b^	4.25 ± 0.87 ^a^	5.86 ± 1.28 ^a^	4.57 ± 0.96 ^a^	3.55 ± 0.77 ^a^	4.94 ± 0.97 ^a^	2.41 ± 0.49 ^a^
*10*	Menthol	1174	1167	0.28 ± 0.08 ^b^	0.05 ± 0.02 ^a^	0.10 ± 0.04 ^a^	tr	tr	0.08 ± 0.03 ^a^	tr
*11*	Isopulegone	1179	1170	0.12 ± 0.05 ^a^	0.64 ± 0.07 ^bcd^	0.76 ± 0.04 ^d^	0.66 ± 0.04 ^d^	0.60 ± 0.07 ^bcd^	0.47 ± 0.06 ^bc^	0.41 ± 0.06 ^b^
*12*	**Pulegone**	1241	1233	**8.55 ± 2.99 ^a^**	**49.90 ± 3.27 ^d^**	**53.44 ± 2.35 ^d^**	**49.70 ± 3.15 ^cd^**	**48.07 ± 3.39 ^cd^**	**34.56 ± 3.36 ^b^**	**37.29 ± 2.97 ^bc^**
*13*	*cis*-Piperitone oxide	1258	1250	1.67 ± 0.20 ^b^	0.35 ± 0.07 ^a^	0.18 ± 0.05 ^a^	0.38 ± 0.06 ^a^	0.15 ± 0.06 ^a^	0.51 ± 0.08 ^a^	0.30 ± 0.08 ^a^
*14*	Isomenthyl acetate	1276	1271	0.07 ± 0.05	-	-	-	-	-	-
*15*	Piperitenone oxide	1370	1366	0.08 ± 0.03 ^a^	0.40 ± 0.07 ^b^	0.48 ± 0.07 ^b^	0.51 ± 0.08 ^b^	0.52 ± 0.08 ^b^	0.30 ± 0.10 ^ab^	0.25 ± 0.06 ^ab^
Sesquiterpenoids										
	*Sesquiterpene hydrocarbons*									
*16*	(*E*)-Caryophyllene	1426	1417	0.11 ± 0.04 ^a^	0.42 ± 0.06 ^ab^	0.85 ± 0.14 ^cd^	0.37 ± 0.04 ^ab^	1.06 ± 0.17 ^d^	0.56 ± 0.08 ^bc^	1.13 ± 0.22 ^d^
*17*	Germacrene D	1488	1484	0.38 ± 0.07 ^a^	1.09 ± 0.11 ^ab^	2.15 ± 0.35 ^bc^	0.85 ± 0.07 ^a^	2.76 ± 0.47 ^c^	1.31 ± 0.19 ^ab^	2.55 ± 0.39 ^c^
*18*	Bicyclogermacrene	1504	1500	0.08 ± 0.03 ^a^	0.32 ± 0.06 ^ab^	0.86 ± 0.18 ^bcd^	0.30 ± 0.05 ^ab^	1.21 ± 0.24 ^d^	0.50 ± 0.10 ^abc^	0.95 ± 0.16 ^cd^
*19*	*trans*-Calamenene	1527	1521	0.09 ± 0.03 ^a^	0.14 ± 0.04 ^a^	0.06 ± 0.02 ^a^	0.12 ± 0.04 ^a^	0.22 ± 0.07 ^ab^	0.38± 0.08 ^b^	0.18 ± 0.04 ^ab^
Grouped components										
	Monoterpene hydrocarbons			28.89	27.91	23.05	26.35	31.91	38.03	41.58
	Oxygenated monoterpenes			70.17	69.81	72.81	70.44	62.35	58.64	53.26
	Sesquiterpene hydrocarbons			0.67	1.97	3.92	1.64	5.25	2.75	4.80
	Others			0.00	0.14	tr	0.07	tr	0.07	tr
	Total identified			99.73	99.83	99.81	98.50	99.53	99.49	99.66

^1^ RI: Experimental linear retention indices relative to C_8_-C_20_ alkanes. ^2^ LI: Literature indices—Adams’ retention indices. ^3^ Relative abundances are given as percentages (mean ± SE) of the total compounds content, *n* = 30; means within rows followed by different superscript letters are significantly different according to Tukey’s HSD test at *p* ≤ 0.05; tr—trace < 0.05%; (-) not detected compounds; the most abundant HS volatiles (recorded in the average content ≥ 15% in at least one treatment) are in boldface.

**Table 2 plants-11-00198-t002:** Standardized coefficients for the first two canonical axes (CA) of variation in 19 *Clinopodium pulegium* HS volatiles from the discriminant functional analysis of seven a priori groups. Significant coefficients are in boldface.

Variables	CA 1	CA 2
(*3Z*)-Hexenol	0.054	0.120
α-Pinene	−0.166	−0.135
Sabinene	**1.023**	**0.871**
ß-Pinene	**−1.516**	**−1.174**
ß-Myrcene	−0.585	−0.575
Limonene	0.230	**−0.925**
Isopulegol	−0.152	0.326
Menthone	**−1.141**	**−0.794**
Isomenthone	**−0.788**	−0.463
Menthol	0.042	−0.086
Isopulegone	0.253	0.237
Pulegone	**−0.600**	**−0.965**
*cis*-Piperitone oxide	−0.178	−0.045
Isomenthyl acetate	−0.262	0.029
Piperitenone oxide	−0.249	0.196
(*E*)-Caryophyllene	**0.878**	**−0.714**
Germacrene D	0.134	0.001
Bicyclogermacrene	−0.457	−0.255
*trans*-Calamenene	−0.291	**0.755**
Eigenvalue	2.950	0.613
% explained variation	70.0%	14.0%

**Table 3 plants-11-00198-t003:** Standardized coefficients for the first two canonical axes (CA) of variation in 18 *Clinopodium pulegium* HS volatiles from the discriminant functional analysis of six a priori groups. Significant coefficients are in boldface.

Variables	CA 1	CA 2
(*3Z*)-Hexenol	−0.019	0.099
α-Pinene	0.257	**1.024**
Sabinene	−0.028	**−1.653**
ß-Pinene	−0.221	−0.136
ß-Myrcene	0.103	**0.710**
Limonene	**0.778**	−0.471
Isopulegol	−0.551	−0.392
Menthone	−0.127	−0.285
Isomenthone	0.327	**0.737**
Menthol	−0.034	−0.264
Isopulegone	−0.121	−0.248
Pulegone	0.550	0.228
*cis*-Piperitone oxide	−0.287	−0.233
Piperitenone oxide	−0.214	0.224
(*E*)-Caryophyllene	**1.194**	**−1.281**
Germacrene D	0.085	0.031
Bicyclogermacrene	−0.159	**0.770**
*trans*-Calamenene	**−0.740**	0.491
Eigenvalue	1.032	0.386
% explained variation	55.0%	21.0%

## Data Availability

Data generated in this research is presented here. Additional details can be obtained from the authors on request.

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
