# Peer review of "Volatile Organic Compound Composition and Glandular Trichome Characteristics of In Vitro Propagated *Clinopodium pulegium* (Rochel) Bräuchler: Effect of Carbon Source"

_plants, 2022, doi:10.3390/plants11020198_

Round 1
Reviewer 1 Report
There authors investigate the impact of different carbon sources on the growth of Clinopodium pulegium. Other than slight changes in volatile production and changes in numbers of glandular trichomes, changes induced by the different carbon sources were minimal. That said, while descriptive, the results were arrived at in a scientific manner. Novelty and significance are factors to consider by the editor.
I would suggest to move figures that effectively display a non-significant result (e.g. Fig. 8) to the supplementary information.
Also, please check that all approproate terms, e.g. in vitro have been italicised throughout.
Author Response
Response to Reviewer 1 Comments
Listed below are the reviewer’s comments and suggestions for authors, followed by authors’ response.
Comment 1: There authors investigate the impact of different carbon sources on the growth of Clinopodium pulegium. Other than slight changes in volatile production and changes in numbers of glandular trichomes, changes induced by the different carbon sources were minimal. That said, while descriptive, the results were arrived at in a scientific manner. Novelty and significance are factors to consider by the editor.
Thank you very much for reading the manuscript. Your comments and suggestions are much appreciated.
Comment 2: I would suggest to move figures that effectively display a non-significant result (e.g. Fig. 8) to the supplementary information.
We believe that Figure 8 displays the results that are not insignificant in the context of our study. Observed increase in density of peltate trichomes, which are the main VOC producers in this species, on glucose-supplemented medium, is rather relevant in terms of VOC production.
Comment 3: Also, please check that all approproate terms, e.g. in vitro have been italicised throughout.
All the appropriate terms have been italicized.
Reviewer 2 Report
This work is related to the study of the influence of different carbon sources in the nutrition medium on VOCs emissions of micropropagated C. pulegium plants, using headspace followed by GC-MS analysis. For this, different conditions were tested: the VOCs were analyzed in relation to the presence, type, and concentration of carbon source in the nutrient medium. The effect of different carbohydrates on the density and size of leaf glandular trichomes were also analyzed. In my opinion, this is an interesting work, well planned and discussed. I have only minor comments that can be found in the attached pdf file.

Author Response
Response to Reviewer 2 Comments
Listed below are the reviewer’s comments and suggestions for authors, followed by authors’ response.
Comment 1: This work is related to the study of the influence of different carbon sources in the nutrition medium on VOCs emissions of micropropagated C. pulegium plants, using headspace followed by GC-MS analysis. For this, different conditions were tested: the VOCs were analyzed in relation to the presence, type, and concentration of carbon source in the nutrient medium. The effect of different carbohydrates on the density and size of leaf glandular trichomes were also analyzed. In my opinion, this is an interesting work, well planned and discussed. I have only minor comments that can be found in the attached pdf file.
Thank you very much for reading the manuscript. Your comments and suggestions are much appreciated.
Comment 2: (Lines 24-26)
The term “we analyzed” was deleted and replaced with “was determined”, according to the reviewer’s suggestion.
Comment 3: (Lines 26-27) Since the authors have not identified their VOCs based on pure standards the term "putatively" identified or " tentatively identified should be used.
“We identified 19 VOCs, primarily belonging to mono- and sesquiterpenes previously described in plants.” has been changed according to the reviewer’s suggestion into “A total of 19 VOCs, primarily belonging to mono- and sesquiterpenes previously described in plants, were tentatively identified.”
Comment 4: (Line 52)
Comma has been inserted, according to the reviewer’s suggestion.
Comment 5: (Line 62)
The term “monoterpene” has been replaced by “monoterpenic”, according to the reviewer’s suggestion.
Comment 6: (Line 93)
The term “in order” has been deleted according to the reviewer’s suggestion.
Comment 7: (Lines 106-107)
The sentence “In addition, we analyzed the effect of different carbohydrates on density and size of leaf glandular trichomes, the main structures involved in the emission of VOCs.“ has been changed according to the reviewer’s suggestion into “In addition, the effect of different carbohydrates on density and size of leaf glandular trichomes, the main structures involved in the emission of VOCs, was analyzed.”
Comment 8: (Line 115 and elsewhere in the text)
Upper case ‘p’ (in P ≤ 0.05) was replaced with lower case letter, according to the reviewer’s suggestion.
Comment 9: (Line129)
Comma has been inserted, according to the reviewer’s suggestion.
Comment 10, Table 1: The authors had not used a chiral column or pure standards to identify if the compound was E or Z or Trans or Cis. These identifications should be deleted and reformulated throughout the manuscript.
It is true that the chiral column was not used. Retention indices of geometric stereoisomers (cis, trans, E and Z) have different values on the non-chiral column, hence these stereoisomers have been identified based on their retention indices. Therefore, we decided not to remove these identifications.
Comment 11: (line 187)
Comma has been inserted, according to the reviewer’s suggestion.
Comment 12: (Lines 214-215) In my opinion, it is important to highlight that sesquiterpenic compounds are related to several health benefits already determined in vitro and in vivo (doi:10.1016/j.indcrop.2012.02.041). So this result, could be interesting regarding this point and should be commented on.
To highlight the importance of sesquiterpenic compounds in relation to health benefits, we added the following sentence: “Sesquiterpenic compounds, which are typically produced in a number of plant species for their antimicrobial and insecticidal properties, as well as their role as insect pheromones [1,37], also display various pharmacological effects and are associated with diverse health benefits [41].”
Although I must admit this addition was made with reluctance, for many good reasons, one being that aromatic and medicinal plants were shown to display various biological activities, much if not all the research being carried out exactly because they provide health benefits known since ancient times, due to the bioactive compounds they produce. Furthermore, pointing out at sesquiterpenes contained in Clinopodium species, otherwise known for the huge prevalence of monoterpenic compounds, makes little sense to me, especially given the fact that not even the monoterpenes contained were discussed in the health benefit context here, since that was not the purpose of the study.
However, to preserve the context and not diverge from the main idea, this entire paragraph was further rearranged into:
Carbohydrates affected the ratio between the two monoterpene fractions, with sucrose exerting the most prominent effect (Table 1). Compared to control plants, the relative percentage of pulegone in plants cultured in the presence of carbohydrates was inversely correlated with that of menthone and isomenthone. Furthermore, higher sesquiterpene fraction (lower monoterpene-to-sesquiterpene ratio) was also observed in C. pulegium plants cultured in the presence of carbohydrates, especially at higher sugar concentrations, compared to control plants (Figure 2). Sesquiterpenic compounds, which are typically produced in a number of plant species for their antimicrobial and insecticidal properties, as well as their role as insect pheromones [1,37], also display various pharmacological effects and are associated with diverse health benefits [41].
All subsequent citations (following [41]) were changed accordingly, and so was the reference list.
Comment 13: (Line 226)
The term “our” has been replaced by “this”, according to the reviewer’s suggestion.
Comment 14: (Line 298)
Comma has been inserted, according to the reviewer’s suggestion.
Comment 15: (Line 443)
The term “our” has been replaced by “this”, according to the reviewer’s suggestion.
Comment 16: (Line 488) Please include the number of replicates.
The number of replicates has been included in the subsection 3.4 Headspace isolation, according to the reviewer’s suggestion [“Two replicates, each consisting of at least 15 randomly chosen plants, were used per treatment.”]. Only one injection was possible for the applied head space.
Comment 17: (Line 536)
Comma has been inserted, according to the reviewer’s suggestion.
Comment 18: (Line 541-542)
The sentence “We identified 19 glandular trichome-derived volatiles, primarily belonging to monoterpenes, dominated by pulegone, ß-pinene and menthone.“ has been changed according to the reviewer’s suggestion into “A total of 19 glandular trichome-derived volatiles, primarily belonging to monoterpenes, dominated by pulegone, ß-pinene, and menthone, were identified.”
Comment 19: (line 544)
Comma has been inserted, according to the reviewer’s suggestion.